# Deep Learning Approaches to Automated Video Classification of Upper Limb Tension Test

**DOI:** 10.3390/healthcare9111579

**Published:** 2021-11-18

**Authors:** Wansuk Choi, Seoyoon Heo

**Affiliations:** 1Department of Physical Therapy, International University of Korea, Jinju 52833, Korea; y3korea@gmail.com; 2Department of Occupational Therapy, School of Medical and Health Science, Kyungbok University, Namyangju-si 12051, Korea

**Keywords:** deep structured learning, supervised machine learning, automated feature extraction, Brachial Plexus Tension Tests, rehabilitation medicine, human action recognition

## Abstract

The purpose of this study was to classify ULTT videos through transfer learning with pre-trained deep learning models and compare the performance of the models. We conducted transfer learning by combining a pre-trained convolution neural network (CNN) model into a Python-produced deep learning process. Videos were processed on YouTube and 103,116 frames converted from video clips were analyzed. In the modeling implementation, the process of importing the required modules, performing the necessary data preprocessing for training, defining the model, compiling, model creation, and model fit were applied in sequence. Comparative models were Xception, InceptionV3, DenseNet201, NASNetMobile, DenseNet121, VGG16, VGG19, and ResNet101, and fine tuning was performed. They were trained in a high-performance computing environment, and validation and loss were measured as comparative indicators of performance. Relatively low validation loss and high validation accuracy were obtained from Xception, InceptionV3, and DenseNet201 models, which is evaluated as an excellent model compared with other models. On the other hand, from VGG16, VGG19, and ResNet101, relatively high validation loss and low validation accuracy were obtained compared with other models. There was a narrow range of difference between the validation accuracy and the validation loss of the Xception, InceptionV3, and DensNet201 models. This study suggests that training applied with transfer learning can classify ULTT videos, and that there is a difference in performance between models.

## 1. Introduction

Whereas research into classifying videos using deep-learning approaches has been inclined to be tentative in rehabilitation medicine fields, recent advances in technologies have accelerated research into analyzing overwhelmed video data. Human action recognition has been expected to achieve a more refined and more scientific educational effect in the environment of the recent academic supply, which is described and consumed in images or motion pictures. Video (including images or motion pictures) data are regarded as a spatiotemporal generalization of image data from a traditional neural network’s point of view [1], and all neural network structures for image classification have been naturally extended and discussed to a three-dimensional version beyond two dimensions [2]. The machine learning process is a given for deriving insights or making classifications and predictions. It refers to the way the data fit into a mathematical model [3]. Particularly, machine learning discovers patterns that do not involve human subjective judgment or other possible biases from a large amount of data, having high predictive power [4]. Since the introduction of a video classification method using a dimensional convolutional neural network (CNN) [5,6], a 3D CNN has been applied to large-scale video classification. Interestingly, however, the performance of the 3D CNN was only slightly better than that of the CNN, which classified each frame of a video as a 2D convolution. As a result of this, it was found that important information in video classification is already contained in individual frames, and that information about “movement” in general is not very helpful in classifying videos [7,8]. Therefore, the researchers tried to combine a CNN with a recurrent neural network (RNN) except for a very short video (about 0.5 s) to improve the performance of video recognition, and they achieved excellent results [9,10].

Looking at the review papers on the various methods and algorithms for image or human behavior recognition, we can see in advance that it is very difficult to focus on combining the functions provided by integration of digital transformation (IDT) to increase precision and reduce background noise, and all of these technologies mentioned. Despite its advantages, there are still many limitations in collecting and processing fully dynamic information [10]. An optical flow method can be used to overcome these shortcomings, but it is still not conducive to sports video movements that are captured from different angles. Deep learning-focused behavior recognition methods can handle large datasets, but increasing the efficiency of the system can be considered as a future scope [11].

The key to combining RNN and CNN [12] into one structure is to make all neurons in a convolutional neural network into cyclic units. In other words, it is an extension of the convolutional neural network itself to the cyclic version. Originally, CNNs are more specific feature maps as they ascend from the lower layer to the higher layer, which have excellent performance to distinguish 2D. The RNN is derived from feedforward neural networks, where connections between nodes can represent temporary dynamic behavior by forming directed graphs along a time sequence. RNN can also handle process variable length sequences of inputs using internal state (memory) [13,14]. Models combined to CNN and RNN including ResNet152V2 [15], MobileNetV2 [16], NASNetMobile [17], among others, MobileNetV2 and ResNet50V2 are known to outperform other models for classifying videos such as sports and human behavior.

Deep learning is a technology that instructs computers to perform tasks similar to those performed naturally by the human brain. Based on the research proposed and analyzed on CNN, RNN, LSTM (Long Short-Term Memory), DBN (Deep Belief Network), and GAN (Generative Adversarial Network), which are networks widely used for behavior recognition tasks in previous studies, this study also utilized frameworks for analyzing human activity and behavior is approached to the therapeutic clinical domain [18].

Due to advances in techniques for classifying such videos, research incorporating deep learning into musculoskeletal fields [19] might be of interest. Given the recent interest in musculoskeletal video data collection [20,21], it is predicted that machine learning or deep learning research using these data would subsequently expand to orthopedic or occupational therapy. Especially due to the coronavirus pandemic, many practical classes have been canceled or reduced, and classical physical training methods are usually the only alternative [22], raising complaints from educators and students; learners are still demanding smart educators who can guide musculoskeletal work in non-face-to-face conditions [23].

Most of the clinical skills and technical parts used in the rehabilitation medicine field are apprenticed, which takes a long time and individual deviations are large because students or interns acquire them from a first-person perspective in terms of education. The image information on the posture of each step shown from the third person point of view may be a timely attempt in the era of a new technology.

In Korea, even in the physical therapy curriculum which currently requires actual practice, face-to-face education is being disrupted due to the spread of the COVID-19 pandemic, and most students rely on superficial video recordings or attend online lectures. In such a circumstance, an accurate image/video analysis model would be able to provide students with learning contents based on accuracy close to actual practice.

Analysis of sports performance during physical activity is an important indicator used to improve the performance of players during a match. In this paper, a review study on video-based technology for sports action recognition for building an automated notation analysis system was introduced and the principle was identified [24]. However, in this study, it is more meaningful and an essential process for the practical stage because it is not limited to theory, but actually grafted such research into actual clinical practice.

As one of these solutions, we attempted to build a deep learning model that recognizes Upper Limb Tension Test (ULTT) videos, which is used as one of the evaluation tools for manual therapy. In this process, we tried to compare several transfer learning models to determine which model is the best at ULTT video recognition. In this work, we compared the performance with existing models by fine tuning models such as NASNetMobile, InceptionV3, and Xception, which were newly released but rarely used for video classification. We aimed to find a good model that can recognize Upper Limb Tension Test (ULTT) behavior well.

## 2. Methods

### 2.1. ULTT Clinical Settings

Pain in the neck, shoulders, and upper limbs, which is commonly encountered clinically, is directly related to the nervous systems. Neurological defects or lesions are highly associated with upper arm neurosurgery, and there is an upper limb neurological test (ULTT) designed by Elvey (1980) as a method to clinically identify these signs [25]. In particular, when the adaptive mechanism of the nervous system is damaged, abnormal muscle tension such as contracture, shortening, and myoclonus occurs in the muscle, and joint range of motion is limited, and sensory disturbance occurs. In addition, when peripheral nerve damage occurs, additional disorders such as restriction of muscle activity, edema, blood circulation disorder, and autonomic nervous system dysfunction such as sweating occur concurrently [26]. Pain in the neck, shoulder, and upper extremities that are commonly encountered clinically is directly related to the nervous system. Neurological defects or lesions are highly correlated with the brachial plexus, and the upper limb tension test (ULTT) devised by Elvey (1980) is a clinical method to confirm these signs [27]. This tension test is to create a posture of maximum tension by gradually applying step-by-step tension to various nerve structures up to the maximum range of motion. Upper Limb Tension Test 1 (ULTT1) is test for median nerve, anterior interosseous nerve, and clinician lowers the shoulder of the anterior interosseous nerve and spreads it 110 degrees. The elbows, wrists, and fingers are straight, and the forearms are twisted. Upper Limb Tension Test 2A (ULTT2A) is another type of median nerve bias test, which is a test operation in the order of the musculocutaneous and axillary are lowered and turned lateral to 10 degrees apart. The forearms, wrists, and fingers are straightened, and the forearms prostrate. In this position, the median nerve, musculocutaneous nerve, and axillary nerve are tense. Upper Limb Tension Test 2B (ULTT2B) is a test for radial nerve bias and is a test in order of shoulder girdle depression, elbow extension, medial rotation of the whole arm, wrist, finger, and thumb flexion. Lastly, Upper Limb Tension Test 3 (ULTT3) is a test for ulnar nerve bias, which is a test movement in lower the patients’ shoulders and rotate them inward to 10 degrees apart. The elbows are straightened, the forearms are propped up, and the wrists are bent so that they are tilted toward the left side, and the fingers are also bent. In this posture, the old nerve is tense (Figure 1). In the text, representative pictures were presented, and in the actual video, each therapist was examined in a slightly different posture. For example, one clinician pressed the shoulder to increase nerve tension, whereas another instructed the patient to move the neck flexion to the left or right. The specific implementation procedures of ULTTs performed by clinicians were summarized using existing references [25,28] and presented at each stage through Table 1 of this paper.

### 2.2. Deep Structured Learning Experimental Settings

Deep learning model experiments for video classification and analysis were conducted in Python 3.8, Opencv 4.2, Keras 2.4, and TensorFlow 2.0 environments. In addition, TensorFlow-gpu 2.0 was used to utilize the effective GPU processor in training time. The specifications used in the experiment were Intel^®^ Xeon^®^ Gold 5120 CPU © 2.20 GHz, 180 GB RAM, two NVIDIA Tesla V100-SXM2-32GB, and Windows 10 64 bit operating system was installed.

### 2.3. Video Collection

Two independent external physical therapists who are not included in the study authors carefully selected only the correct data from four type of ULTT videos (ULTT1, ULTT2A, ULTT2B, and ULTT3) on YouTube. Search terms are upper limb tension test, ULTT, upper limb tension test 1, ULTT1, median nerve bias, upper limb tension test 2A, ULTT2A, median nerve bias, upper Limb tension test 2B, ULTT2B, radial nerve bias, upper limb tension test 3, ULTT3, ulnar nerve bias. Each of the four classes consisted of 130 video clips. A Python program for collecting video address, 4k Video Downloader, Fast Duplicate File Finder, Free Video to JPG Converter and Windows photo editing program were used in the video image download, editing, and frame extraction process.

### 2.4. Dataset and Preprocessing

An image was extracted every frame from this video, and each image was resized to (224, 224, 3) for useful batch learning. In addition, the data augmentation technique was applied to reduce generalization loss and to make the model more powerful. Each class of the dataset is ULTT1, ULTT2A, ULTT2B, and ULTT3, and consists of 131,458 images. The process of moving and examining joints in the ULTT image frame was analyzed, and frames including actions and postures unnecessary for training were removed. In the entire ULTT images dataset of 103,116 frames, the ratio of training and validation frames was assigned to 8:2 (Figure 2 and Figure 3).

### 2.5. Working with the Dataset

#### 2.5.1. Extracting Features from the Frames Using CNN

ULTT video is one of the physiotherapy techniques for examining a patient, and it is difficult to analyze the image in engineering due to the surrounding environment, image quality, camera angle, and various gestures. Therefore, since it is difficult to estimate motion features with existing methods, we tried to estimate accurate ULTT motions using a deep learning-based classification method.

CNN (Convolutional Neural Network) is a technology that performs two-dimensional image analysis and classification prediction based on an artificial neural network model in the field of computer vision. In this study, we also tested ULTT motion estimation using CNN. Figure 4 shows the fundamental model structure for learning the ULTT dataset, and each fine-tuning model was trained. In the process of the experiment, we tried to achieve the best training result by designing a dense layer suitable for dataset learning through trial and error and setting hyperparameters.

The pretrained CNN models DenseNet121, DenseNet201, InceptionV3, NASNetMobile, ResNet101, VGG16, VGG19, and Xception provided by Keras was used to extract feature vectors from the ULTT videos. Using the weights of these models pre-trained on the ImageNet dataset consisting of 1000 classes (1.2 million labeled images) contributed to saving a lot of time and effort to obtain results (Figure 4 and Figure 5).

We derived the classification probability $cost_{i}$ by adding a density layer at the end, as shown in the following Equations (1) and (2) for probabilistic calculation with 4 types of classes which could lead to minimize loss functions.
(1)$$cost_{i}=\frac{1}{m}\overset{m}{\underset{i=0}{\sum }}\left(\mathrm{dense}\left(model\left(I_{i}\right)\right)-y_{i}\right)$$
(2)$$\mathrm{dense}\left(I_{i}\right)=\mathrm{ReLU}\left(b_{j}+\overset{n}{\underset{i=0}{\sum }}I_{i}\times w_{ij}\right)$$

Here, $I_{i}$ is the training image, model $\left(I_{i}\right)$ is each transfer learning model, and $y_{i}$ is the predicted classification probability (ground truth or label). In this training, a learning rate of 0.0001, batch size of 32, and epochs of 100 were used by fine-tuning hyperparameters.

#### 2.5.2. Loss Function

The loss function is the ‘difference or error between the predicted value and the correct value’ and was used to optimize the neural network as a function that was minimized by the selected optimizer. In this process, multiple classification is performed by supervised learning, and the correct answer class is expressed as a label value (e.g., 0, 1, 2, 3, …) rather than a ‘one-hot encoding’, thus the ‘sparse_cartegorical_crossentropy’ loss function was used. Additionally, since it is a multiple classification, ‘Softmax’ (Figure 6) was used as the activation function of the output layer (Equation (3)).
(3)$$f{\left(s\right)}_{i}=\frac{e^{s_{i}}}{\sum_{j}^{c}\;e^{s_{j}}}\;CE=-\overset{c}{\underset{i}{\sum }}t_{i}\log(f{\left(s\right)}_{i})$$

#### 2.5.3. Saving the Best Model and Classifying Videos

The authors saved the model weights for the iteration whenever the validation loss was the smallest, because saving the model only with the best accuracy, regardless of the loss, would not result in the best model. Therefore, for this study, the smallest loss was used to specify the best model. Then, we defined the same architecture we defined as training to classify the video. Next, we loaded the model weights, and then we were ready to classify the videos. This involved obtaining the video, extracting the frames, and then using the CNN model to extract features from those videos by passing these features to the architecture.

## 3. Results

The results showed almost similar progress; training loss and validation loss showed a tendency to increase in value from the Xception to the ResNet101, whereas training account and validation account showed a decreasing direction. Validation loss and validation accuracy of Xception, inceptionV3, and DenseNet201, which are evaluated as models with excellent performance, are similar between models, but there is a very small difference in validation loss at three decimal places and validation accuracy at four decimal places. The NASNetMobile and DeseNet121 models showed intermediate performance, and the validation loss of the NASNetMobile model was slightly lower and the validation accuracy was slightly higher. The validation accuracy of VGG19 and ResNet101, which showed low performance, were similar to each other, but the validation loss was significantly lower in VGG19 (Table 2, Figure 6, Figure 7 and Figure 8).

## 4. Discussion

ULTT is an upper limb tension test and a representative evaluation method that can clinically identify nerve muscle shortening and stiffness [29]. In particular, when the adaptation mechanism of the nervous system is damaged, abnormal muscle tension such as contracture, shortening, and myoclonus occurs in the muscle, and limitation of the joint range of motion and sensory disturbance also occur. In addition, when peripheral nerve damage occurs, additional disorders such as restriction of muscle activity, edema, blood circulation disorder, and autonomic nervous system dysfunction such as sweating occur concurrently. Pain in the neck, shoulder, and upper extremities commonly encountered clinically is directly related to the nervous system. Neurological deficits or lesions have many associations with the brachial plexus, and ULTT was adopted as a clinical method for identifying these signs [30]. ULTT, which is evaluated as a simple yet clinical core technique [31], was used to see that it can be implemented in a therapeutic environment beyond the realm of setting the realm of human action recognition in sports or motion itself. These attempts have not been actively attempted in other papers, and are expected to be used to maximize communication between medical staff in education and clinical settings.

In this study, the performance of eight types of models that classify four types of ULTT video through transfer learning was compared. During data training, as the number of epochs increased, training and validation accuracy increased while training and validation loss decreased. For models with good performance such as Xception, InceptionV3, and DensNet201, the accuracy increased sharply between 20–30 epochs and the loss decreased sharply. However, in the case of models such as VGG16, VGG19, and ResNet101, the degree of increase in accuracy or decrease in loss was relatively modest. Moreover, in the case of the excellent model, validation and training loss became constant after 10 epochs, and validation and training accuracy of 97–99% were obtained.

Compared with this study, the validation loss (error) of the study classifying sports videos using transfer learning was lower in the VGG16 model than in the VGG19 [32]. It would be presumed that these different results were probably due to differences in datasets and parameters. Meanwhile, another study that categorizes various sports (18 types) using the VGG16 transfer learning model had a validation accuracy of 0.93 and a validation loss of less than 0.25, indicating that the validation accuracy was low and the validation error was higher than the results of our study [33]. This suggests that the performance of VGG16 applied to the dataset of this study would be surpassed.

In addition, the validation loss of ResNet101, VGG16, and VGG19 decreased gently as the epochs increased, whereas the validation loss values of other pre-trained models including Xception mostly fell to the shortest before 20 epochs. The validation loss values of ResNet101, VGG16, and VGG19 of these three models were relatively higher than those of other learning models. These results suggest that DensNet121, DensNet201, Xception, InceptionV3, and NASNetMobile models are suitable and Sophisticated models for the dataset from this study. In addition, the validation loss of the ResNet101 model was the highest among the models with a gentle fall, and the validation accuracy increased unstable as the process progressed, and the value was also the lowest.

Xception and InceptionV3 showed the best performance among eight models. Xception is further derived from the IncetionV3 model and obtained the highest performance (numerical) among the models. The InceptionV3 model uses a method to simultaneously map each cross-channel correlation and spatial correlation by adding a channel dimension to the convolutional layer as shown in Figure 7. For example, different filter results are obtained at each step for features such as human eyes, nose, mouth, arms, and joints, and classification probability is estimated based on this. Certain types of deep learning models should learn well by increasing the depth of the hidden layer, but in the case of a hidden layer that is too deep, there may be problems such as vanishing/exploding gradients, etc. Therefore, the Xception model developed to solve this problem solved the learning problem by preventing vanishing/exploding gradients by adding the previous identity data through the residual connection method. In the ULTT dataset, information analysis on the direction of the arm and joints is important, and our team obtained interesting and satisfactory results as the Xception model performed very well in classifying and analyzing each piece of delicate feature information through cross-correlation.

Unexpectedly, ResNet101 has performed favorably in studies for human action recognition [34,35]. In this study, ResNet101 model performed significantly lower compared with other research, due to the fact that it is estimated that the noise of background information in the ULTT dataset interferes with its training. In addition, models other than ResNet101 applied in this trial presented advantageous performance on the ULTT dataset; however, when the validation accuracy approaches 1.0 and the validation error approaches 0.0, it would be determined that they could be analyzed as generalization errors.

Deep learning-based solutions for computer vision have made it easier to deliver technical content for educational purposes or clinical use. It has been mentioned in other papers that conventional image data contain many hidden pieces of information and patterns that can be used for human activity recognition (HAR), and HAR can be applied to many areas such as behavioral analysis, intelligent video surveillance, and robot vision [36]. In terms of erroneous classification, existing hand-engineered and machine learning-based solutions have little or no ability to handle overlapping tasks. A fine-tuned pretrained CNN model learns spatial relationships at the frame level [11,18,37]. In this paper, in order to derive an optimized method, the video classification performance was compared with eight transfer learning models (Xception, inceptionV3, DenseNet201, NASANetMobile, DensNet121, VGG16, VGG19, ResNet101) that are most reliable at the current time point. It is believed that the aforementioned loss could be greatly reduced or corrected by repeating their verification accuracy and loss testing.

Other studies have used so-called HAR, which automatically identifies the physical activity that humans perform, to infer the behavior of one or more people from a series of observations captured by sensors, video, or still images [38]. Recognizing human action in a video sequence is a very difficult task due to issues such as background clutter, partial occlusion, changes in scale, viewpoint, lighting, and shape. In this study, to improve these shortcomings, CNN models are actively used to show improved performance compared with some existing deep learning-based models.

Although this study presented approximately satisfactory results, it has several limitations. The labeling of the dataset constructed in this paper is for the purpose of simple classification. Therefore, since the classification of the four types of ULTT data set is estimated as a probability value between 0 and 1, more accurate classification performance was obtained in ResNet101, Dense121, etc., which were used before the recently published model.

Despite many efforts, the test accuracy scores in this study seemed to be not very significant. Our model did not generalize to the test set because it could not accurately predict the first data it presented. Our technical team analyzed several possible causes of lowering test accuracy. First, data leakage was not the cause of the lower test accuracy. The training, validation, and test sets were randomly sampled, and the test set was divided, which may lead to data leakage. However, we decided that data leakage was not the cause of lowering test accuracy, as we aimed to make it similar to the data we had never seen before in the light of motion from humans. Second, it is possible that overfitting had occurred. This is according to the more complex the model, the more accurate it can predict on the training data, but when quite complex it becomes too sensitive to each data point in the training set and does not generalize well to new data. Third, we depended only on pretrained models such as Xception for video classification. If libraries such as OpenPose and Keypoint Detection were used, it is expected that the test accuracy score would be higher by analyzing the joint movement more accurately.

Deep learning technology does not mean better creation of missing information. More precise prognosis could be predicted only when the various factors that affect the patient’s prognosis [39] are collected more closely. There would be a limitation to learning sufficient prior knowledge from various input data [40]. A method of giving prior knowledge to a deep learning model has also been proposed, but this is to inform the model with prior knowledge, and ultimately there may still be a problem in which the model cannot be trained from data, and the prior knowledge given in this way is renewed for every problem. There may be issues that need to be defined.

Patients’ various clinical information must be highly accurate and data-driven in a variety of ways to enable the realization of true Precision Medicine [41]. Deep learning technologies in this research would not perform efficiently compared with other machine learning techniques as Artificial neural network (ANN), Support Vector Machine (SVM), Logistic Regression, Multi-Layer Perceptron (MLP), and Random Forest.

There is a limitation to the technical transfer that takes place within the institution to which apprentices or novice clinicians belong when they are in the training process or upgrading their skills. It is considered that educational advancement in posture and clinical technology through various images and image analysis could be introduced through this study.

## 5. Conclusions

The systematic design and modeling of sequential deep learning for human action recognition is expected to be a tool to help with big data processing and educational purposes, which are important in the current academic communication methods described in images. Generally, clinical trials in the medical field using deep learning mechanisms have a fundamental purpose in making accurate predictions of patients’ prognosis or in making judgments for accurate diagnosis [23,26]. In this study, in a similar context, it was designed to provide a scientific basis for technical judgment and clinical decision-making processes that should be applied differently depending on the patient’s condition.

Sufficient numbers and quality data are needed for machine learning techniques to show stronger performance than statistical techniques [40], according to the fact that most clinical metadata could be numerous and unrefined. High-quality cohort data may be needed to fully utilize the strengths of deep learning [41].

In this study, the performance of several pretrained models, including Xception, to classify ULTT video was compared. Unfortunately, despite obtaining high validation accuracy and low validation loss scores in most models except for ResNet101, the test accuracy scores may seem insufficient in the degree of completion compared with other similar studies. Although we judged that the dataset has class balance and only presented accuracies, in the next study, precision, recall, and F1 score will be additionally presented so that model performance can be evaluated in more detail. Additionally, as a study to analyze classification problems, we will be able to present a confusion matrix together. Additionally, we will try to find the model at the optimal point where the generalization performance is at maximum. If learning using an advanced library (OpenPose, Keypoint Detection, etc.) and advanced dataset composing key point labels or line labels is attempted, more improved video classification results and image analysis results will be obtained. Furthermore, follow-up research could apply the recently introduced Full Convolution Dense Dilated Network (FCdDN) model, which achieves a favorable segmentation efficiency performance while ensuring high accuracy [42]. We aim to explore more diverse ULTT datasets in order to improve the model’s performance. Once this method is applied, it would be possible to promote the convenience of learners and teachers by recognizing various treatment and examination movements.

## Figures and Tables

**Figure 1 healthcare-09-01579-f001:**
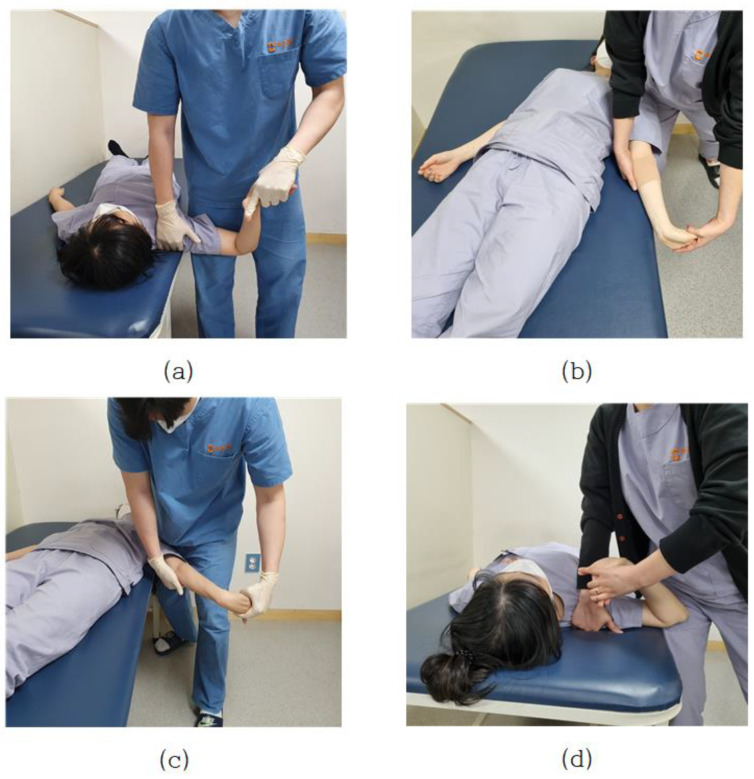
(**a**) Upper Limb Tension Test 1 (ULTT1); (**b**) Upper Limb Tension Test 2A (ULTT2A); (**c**) Upper Limb Tension Test 2B (ULTT2B); (**d**) Upper Limb Tension Test 3 (ULTT3).

**Figure 2 healthcare-09-01579-f002:**
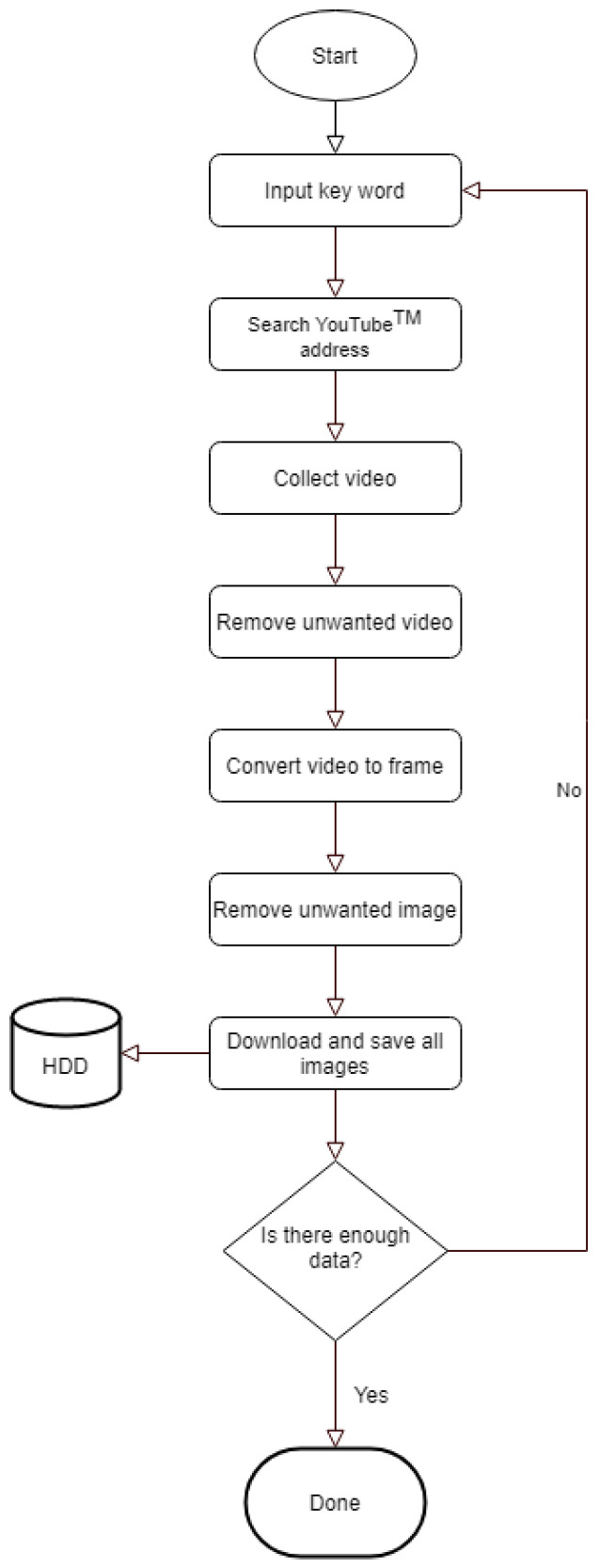
Image dataset collection process.

**Figure 3 healthcare-09-01579-f003:**
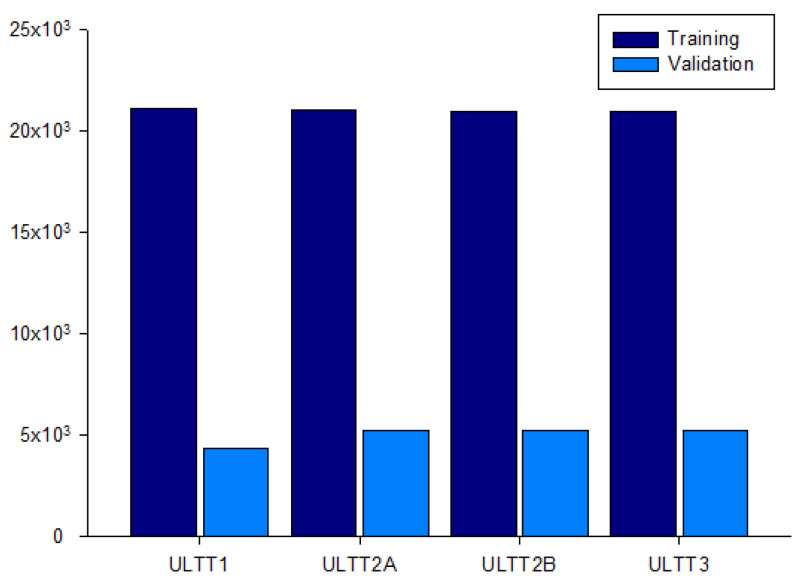
Dataset for the distribution of 103,116 ULTT images.

**Figure 4 healthcare-09-01579-f004:**
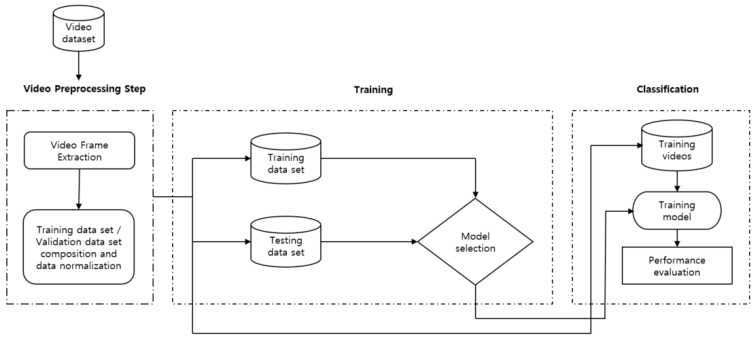
Proposed framework for human action recognition.

**Figure 5 healthcare-09-01579-f005:**
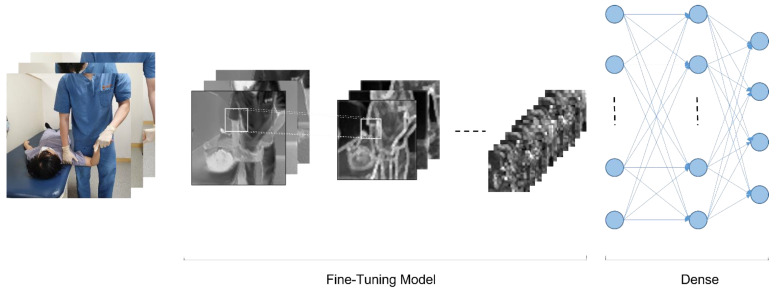
Basic model structure for video classifications.

**Figure 6 healthcare-09-01579-f006:**
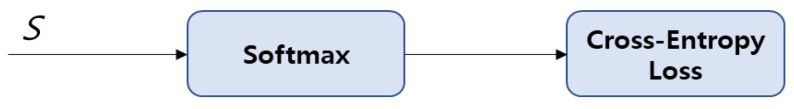
Flow diagram of activating function from ‘Softmax’.

**Figure 7 healthcare-09-01579-f007:**
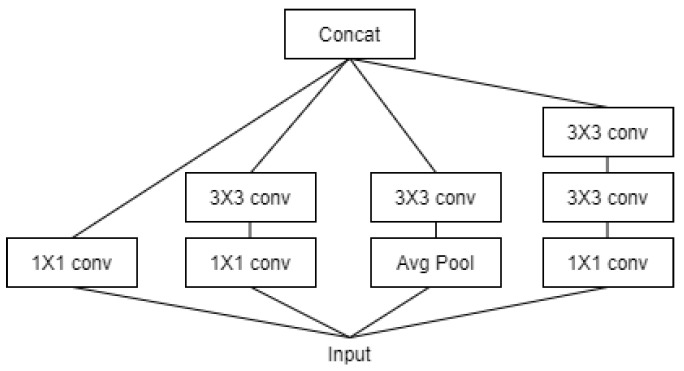
Architecture of Inception V3 Model.

**Figure 8 healthcare-09-01579-f008:**
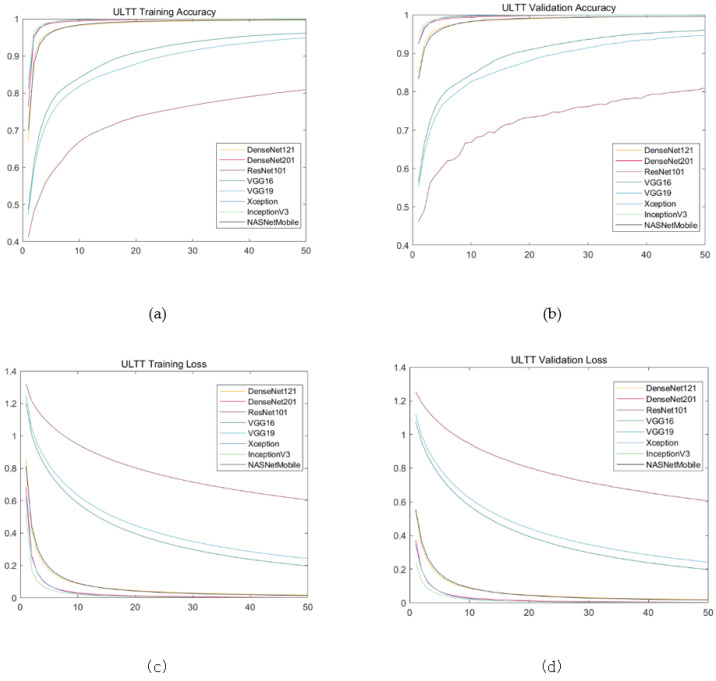
Training results on ULTT data set. (**a**) Training accuracy; (**b**) validation accuracy; (**c**) training loss; (**d**) validation loss.

**Table 1 healthcare-09-01579-t001:** Detailed Process of Upper limb tension tests (ULTTs).

ULTT 1	ULTT2A	ULTT2B	ULTT3
Starting position. Note patient’s thumb and fingertips supported, plus some of the weight of the arm taken on the therapist’s thigh. Shoulder abduction to symptom onset, or tissue tightness, or approximately 100 degrees. Wrist extension. Make sure the shoulder position is kept stable. Wrist supination, again making sure that the shoulder position is kept stable. Shoulder lateral rotation, to symptom onset or where the tissues tighten a little. Elbow extension to symptom onset. Neck lateral flexion away, making sure it is whole neck and not just the upper cervical spine. Neck lateral flexion towards. This should ease evoked symptoms.	Patient has her shoulder girdle just over the side of the bed Shoulder girdle depression (via the therapist’s thigh) to symptoms or where the tissues tighten a little Elbow extension Whole arm lateral rotation, keeping shoulder girdle depressed Wrist and finger extension (note suggest grip in the inset)	The patient lies with their shoulder just over the side of the bed, the therapist uses his thigh to carefully depress the shoulder girdle Elbow extension Notice how the therapist has brought his left arm ‘around’ to grasp the patient’s wrist in order to medially rotate the whole arm Whole arm medial (internal) rotation Wrist and thumb flexion can be added. Leave the fingers out as the extensors will be too tight Adding a few degrees of shoulder abduction will sensitize the test and elevation of shoulder girdle will provide structural differentiation	Starting position—let patient’s elbow rests on the therapist’s hip Wrist and finger extension, ensure 4th and 5th fingers are extended Pronation Shoulder lateral rotation, ensuring wrist position is maintain Elbow flexion Block shoulder girdle elevation by pushing first into the table

**Table 2 healthcare-09-01579-t002:** Model Performance comparison for transfer learning from ULTT datasets.

Model	Training Loss	Training Accuracy	Validation Loss	Validation Accuracy
Xception	0.0012	0.9999	0.0014	0.9999
InceptionV3	0.0016	0.9998	0.0024	0.9996
DenseNet201	0.0037	0.9998	0.0033	0.9996
NASNetMobile	0.0151	0.9977	0.0173	0.9967
DenseNet121	0.0181	0.9972	0.0197	0.9965
VGG16	0.1962	0.9619	0.1973	0.9605
VGG19	0.242	0.9491	0.2418	0.9467
ResNet101	0.6044	0.8093	0.6053	0.8102

## Data Availability

Data available from the corresponding author S.H. on request.

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
