# Peer review of "Deep Learning Approaches to Automated Video Classification of Upper Limb Tension Test"

_healthcare, 2021, doi:10.3390/healthcare9111579_

Round 1

Reviewer 1 Report

The authors have responded to most of our comments but there are still some that require addressing.

  • Discussion Section: It should add more discussion about the obtained results in Table 2 and Figure 7.
  • Conclusion Section: First, the word count for this section should be reduced. Second, the authors should explain precisely what they conclude from this study. This section is not clear. The authors did not respond to this comment.
  • English writing: This paper needs minor proofreading of the entirety of the paper to remove all the issues related to typos, spelling, and grammar mistakes.
  • List of references: References need order in the text, reference [37] is not used in the text. Some references do not contain enough information such as [7], [8] … etc. This paper needs a minor check of the reference list. The authors did not respond to this comment.

Author Response

The authors have responded to most of our comments but there are still some that require addressing.

  • Discussion Section: It should add more discussion about the obtained results in Table 2 and Figure 7.

: About 740 words of discussion have been added to the review, and the contents are as follows.

[Compared to this study, the validation loss (error) of the study classifying sports videos using transfer learning was lower in the VGG16 model than in the VGG19 model (Muhammad Rafiq 2020). It would be presumed that these different results were probably due to differences in data sets and parameters. Meanwhile, another study that categorizes various sports (18 types) using the VGG16 transfer learning model had a validation accuracy of 0.93 and a validation loss of less than 0.25, indicating that the validation accuracy was low and the validation error was higher than the results of our study (Mohammad Yasir Farhad 2020). This suggests that the performance of VGG16 applied to the data set of this study would be surpassed.

In addition, the validation loss of ResNet101, VGG16, and VGG19 decreased gently as the epochs increased, whereas the validation loss values of other pre-trained models including Xception mostly fell to the shortest before 20 epochs. The validation loss values of ResNet101, VGG16, and VGG19 of these three models were relatively higher than those of other learning models. These results suggest that DensNet121, DensNet201, Xception, InceptionV3, and NASNetMobile models are suitable and Sophisticated models for the dataset from this study. In addition, the validation loss of the ResNet101 model was the highest among the models with a gentle fall, and the validation accuracy increased unstable as the process progressed, and the value was also the lowest.

Muhammad Rafiq 1, Ghazala Rafiq 1, Rockson Agyeman 1, Seong-Il Jin 2, Gyu Sang Choi 1. Scene Classification for Sports Video Summarization Using Transfer Learning. Sensors (Basel). 2020 Mar 18;20(6):1702. doi: 10.3390/s20061702.

Mohammad Yasir Farhad, Shahadat Hossain , MD. Rezaul Karim Tanvir, and Shayhan Ameen Chowdhury. Sports-Net18: Various Sports Classification using Transfer Learning. 2020 2nd International Conference on Sustainable Technologies for Industry 4.0 (STI), 19-20 December, Dhaka

Although this study presented approximately satisfactory results, it has several limitations. The labeling of the data set constructed in this paper is for the purpose of simple classification. Therefore, since the classification of the four types of ULTT data set is estimated as a probability value between 0 and 1, more accurate classification performance was obtained in ResNet101, Dense121, etc., which were used previously than the recently published model.

Despite many efforts, the test accuracy scores in this study seemed to be no very significant. Our model did not generalize to the test set because it could not accurately predict the first data it presented. Our technical team analyzed several possible causes of lowering test accuracy. First, data leakage was not the cause of the lower test accuracy. The training, validation, and test sets were randomly sampled, and the test set was divided, which may lead to data leakage. However, we decided that data leakage was not the cause of lowering test accuracy, as we aimed to make it similar to the data we had never seen before in the light of motion from human. Second, it is possible that overfitting has occurred. This is according to the more complex the model, the more accurate it can predict on the training data, but quite complex it becomes too sensitive to each data point in the training set and does not generalize well to new data. Third, we depended only on pretrained models such as Xception for video classification. If libraries such as OpenPose and Keypoint Detection were used, it is expected that the test accuracy score would be higher by analyzing the joint movement more accurately.]

  • Conclusion Section: First, the word count for this section should be reduced. Second, the authors should explain precisely what they conclude from this study. This section is not clear. The authors did not respond to this comment.

: We reduced the content as you said, and instead, we placed the conclusion in the last paragraph in the conclusion section. The contents are as follows.

[In this study, the performance of several pretrained models including Xception to classify ULTT video was compared. Unfortunately, despite obtaining high validation accuracy and low validation loss scores in most models except for ResNet101, the test accuracy scores may seem insufficient in the degree of completion compared to other similar studies. Although we judged that the data set has class balance and presented only accuracy, in the next study, precision, recall, and F1-score will be additionally presented so that model performance could be evaluated in more detail. Also, as a study to analyze classification problems, we will be able to present a confusion matrix together. And we will try to find the model at the optimal point where the generalization performance is maximum. If learning using advanced library (OpenPose, Keypoint Detection etc.) and advanced data set composing key point label or line label is attempted, more improved video classification results and image analysis results will be obtained.]

  • English writing: This paper needs minor proofreading of the entirety of the paper to remove all the issues related to typos, spelling, and grammar mistakes.

: We have reviewed by English Language editors in this draft.

  • List of references: References need order in the text, reference [37] is not used in the text. Some references do not contain enough information such as [7], [8] … This paper needs a minor check of the reference list. The authors did not respond to this comment.

: We fixed mentioned issues in the paper precisely.

Reviewer 2 Report

It is necessary for the authors to respond to both reviewers, and to highlight in red in the manuscript the changes made.

Author Response

All parts of the reviews raised again have been revised or removed, and these parts are written in red letters. Thank you.

Reviewer 3 Report

The abstract doesn't present the context & goal of the paper and is focused only on the technical aspects of the research. The beginning of the abstract "It is demanded to present..." is rather steep and should be rephrases / preceded by the presentation of the general goal of the paper.

In the introduction:

  • "It is used in several fields [4]." construction is too vague. 
  • "research incorporating deep learning into musculoskeletal fields [20] will accelerate." - how can you "predict" the acceleration of the research in the future? This statement is valid only based on an analysis of the evolution of the number of papers in the field over the last few years, not based on one single paper given as reference.

In the "Results" chapter:

  • "[the transfer models] are in almost the same order, and the validation loss of the previous model was low, and the validation accuracy was high." - are...how in the same order? No obvious meaning stated

The paper should be proofread as the phrasing sounds often unnatural and includes grammar and/or spelling mistakes. These aspects make it very difficult (sometimes impossible) to read and follow.

e.g.

  • Video (include images or motion pictures) data are regarded - including, is regarded
  • "Upper Limb Tension Test (ULTT) is for most patients, the normal adaptation mechanism of the nervous system is disrupted or disrupted after damage to the nervous system, so they complain of severe functional deficits and pain [26]." - major editing and rephrasing needed
  • Deep Structured Laerning Experimental Settings - "learning"
  • 4k Vido Downloader - should be "Video"
  • 2.5.3 - The authors saved the model weights for that iteration whenever the validation loss was the samllest - shoul de "smallest"
  • Figure 2 - adress instead of address
  • etc.

Author Response

Response to Reviewer 3’s Comments

Point 1: The abstract doesn't present the context & goal of the paper and is focused only on the technical aspects of the research. The beginning of the abstract "It is demanded to present..." is rather steep and should be rephrases / preceded by the presentation of the general goal of the paper.

Response 1: We revised it as below.

The purpose of this study was to classify ULTT videos through transfer learning with pre-trained deep learning models and compare the performance of the models.

Point 2: In the introduction:

"It is used in several fields [4]." construction is too vague.

"research incorporating deep learning into musculoskeletal fields [20] will accelerate." - how can you "predict" the acceleration of the research in the future? This statement is valid only based on an analysis of the evolution of the number of papers in the field over the last few years, not based on one single paper given as reference.

Response 2: Thank you for your accurate criticism, we revised it as below.

-"It is used in several fields (deleted)

-Due to advances in techniques for classifying such videos, research incorporating deep learning into musculoskeletal fields [19] might be interested.

Point 3: In the "Results" chapter:

"[the transfer models] are in almost the same order, and the validation loss of the previous model was low, and the validation accuracy was high." - are...how in the same order? No obvious meaning stated

Response 3: Thank you for pointing out the conclusion, and I revised and supplemented it with the sentences below. I'm sorry for the confusion caused by incorrect sentences.

-The results showed almost similar progress; training loss and validation loss showed a tendency to increase in value from the Xception to the ResNet101, while training account and validation account showed a decreasing direction.

Point 4: The paper should be proofread as the phrasing sounds often unnatural and includes grammar and/or spelling mistakes. These aspects make it very difficult (sometimes impossible) to read and follow.

e.g.

Video (include images or motion pictures) data are regarded - including, is regarded

"Upper Limb Tension Test (ULTT) is for most patients, the normal adaptation mechanism of the nervous system is disrupted or disrupted after damage to the nervous system, so they complain of severe functional deficits and pain [26]." - major editing and rephrasing needed

Response 4:

-Pain in the neck, shoulders, and upper limbs, which are commonly encountered clinically, is directly related to the nervous systems. Neurological defects or lesions are highly associated with upper arm neurosurgery, and there is an upper limb neurological test (ULTT) designed by Elvey (1980) as a method to clinically identify these signs.

Deep Structured Laerning Experimental Settings - "learning"

4k Vido Downloader - should be "Video"

  • Fixed in each point.

2.5.3 - The authors saved the model weights for that iteration whenever the validation loss was the samllest - shoul de "smallest"

-           Fixed in each point.

Figure 2 - adress instead of address

etc. -      Fixed in each point and others

Thank you very much for all the support.

Round 2

Reviewer 2 Report

There have been some improvements, but the manuscript cannot be published as is.
- To enrich its bibliography, the authors should also present a new architecture, called FCdDNet [1], which is only a few months old, and say how segmentation can add value to their approach:
[1] Ouahabi, A.; Taleb-Ahmed, A. Deep learning for real-time semantic segmentation: Application in ultrasound imaging. Pattern Recognit. Lett. 2021, 144, 27-34. 
- In expressions(1), (2) and (3) the parameters are not defined and neither is the "*" symbol.
 This is unacceptable in a journal like Healthcare.

- It is surprising that the performance of ResNet101 (Table 2) can be so bad. Please check this and justify your results.
Some typos remain, for example instead of "eg" write "e.g.".

Author Response

Thank you very much for systematically and carefully analyzing the results of our triviality experiments and attempts.

Reviewer 3 Report

Major changes have been made by the authors, as suggested in my first review.

However there are still some unnatural constructions which should have been rephrased / removed completely, not moved in a different part of the paper.

E.g. "Most of the patients that medical staff meet in the clinical setting, the normal adaptation mechanism of the nervous system is disrupted or disrupted after damage to the nervous system, so they complain of pain along with serious functional deficits" - this illogical construction was correctly rephrased in the "Methods" chapter, yet it appears now in the beginning of the Discussion section, saying more or less the same things, but with different words. I think there is a mistake and that part should probably be removed from the Discussion (or simply summarized in a few words/sentences).

I have no other major observations.

Author Response

Thank you very much for systematically and carefully analyzing the results of our triviality experiments and attempts.

This manuscript is a resubmission of an earlier submission. The following is a list of the peer review reports and author responses from that submission.

Round 1

Reviewer 1 Report

This paper presents a model to classify the videos for the upper limb tension test. The model compares different deep learning models that could inform fine tune the transfer model to classify the four main movements from the test. The novelty of the paper is very low and the study is conducted in a sloppy way, lack of details that can justify the validity of the work. The way how "deep learning" is applied for this scenario is not interesting to readers.

Reviewer 2 Report

The paper entitled "Deep Learning Approaches to Automated Video Classification of Upper Limb Tension Test" is interesting and original, however some improvements are needed.

- Among the Deep Learning architectures compared, the authors should also introduce a new architecture, called FCdDNet [1], which is only four months old, and say how it could add value to their approach :

[1] Ouahabi, A.; Taleb-Ahmed, A. Deep learning for real-time semantic segmentation: Application in ultrasound imaging. Pattern Recognit. Lett. 2021144, 27–34. 

- The expression (1) lacks justification

- In expression (2), what does the symbol "*" mean?

- There are several possible loss functions (see e.g. reference [1]), the authors do not mention the loss function used.

- It is surprising that the performance of ResNet101 can be so bad. Please check this.

- Finally, the conclusion is confusing, and the reader loses the very purpose of the work. Please clearly state the conclusion, redefine the objective of the work and quantify the contribution of the proposed method as well as its interest.

Reviewer 3 Report

The authors have to address all concerns carefully to ensure that all issues affecting the sobriety of the paper are eliminated.

  • Keywords: We suggest that the authors should replace keywords such as “deep learning” and “upper limb tension test" because these keywords are already found in the paper title. It is preferable that they replace them with other words in order to expand the reach of the paper.
  • Methodology Section: It needs some improvement to be clearer. For example, there should be a distinction between sub-headings 2.1 and 2.2.
  • Could the authors add realistic scenarios/examples showing the usefulness of their study?
  • Discussion Section: The authors provide an appropriate discussion as well as the limitations in the last paragraph (which is good). It is preferable to add more discussion about the obtained results in Table 1 and Figure 6.
  • Conclusion Section: First, the word count for this section should be reduced. Second, the authors should explain precisely what they conclude from this study. This section is not clear.
  • Figures: Some Figures require redrawing to be clearer (some words/lines are blurry) such as [2], [5] and [6]
  • Paraphrase text: There are many sentences taken from previous papers (this is not acceptable). Authors have to paraphrase all sentences and paragraphs taken from the previous papers. The authors must avoid taking whole sentences from the original papers even if it is for the same authors. They are obligated to paraphrase during the revision of the entire paper. Plagiarism is not acceptable at all (please check the entire paper):
  • “shoulder girdle depression, shoulder abduction, shoulder external rotation, forearm supination, wrist and finger extension, elbow extension, and cervical side flexion”, “shoulder girdle depression, elbow extension, lateral rotation of the whole arm, wrist, finger, and thumb extension. Upper Limb Tension Test 2B (ULTT2B) “ and “shoulder girdle depression, shoulder abduction, shoulder external rotation, wrist and finger extension, elbow flexion, and shoulder abduction” (page2) … etc.

.

.

.

Etc.

  • English writing: This paper needs moderate proofreading of the entirety of the paper to remove all the issues related to typos, spelling, and grammar mistakes. Authors must remove all grammatical and typo problems to improve the quality of English writing and to make this paper mistakes-free.
  • List of references: References are recent, sufficient and relevant to the paper topic. However, references need order in the text, for example, the reference [26] is used before the references [24] and [25]. Some references do not contain enough information such as [7], [8] … etc. This paper needs a minor check of the reference list.